

# Unambiguous simulation of diffusive charge transport in disordered nanoribbons

Henrique Pina Veiga[1,2*], Simão Meneses João[1,3], João Manuel Alendouro Pinho[1],
João Pedro Santos Pires[1] and João Manuel Viana Parente Lopes[1†]

**1** Centro de Física das Universidades do Minho e do Porto (CF-UM-UP) and
Laboratório de Física para Materiais e Tecnologias Emergentes LaPMET,
University of Porto, 4169-007 Porto, Portugal
**2** Institute for Systems and Computer Engineering,
Technology and Science (INESC TEC), 4150-179 Porto, Portugal
**3** Department of Materials, Imperial College London,
South Kensington Campus, London SW7 2AZ, United Kingdom

★ up201805202@edu.fc.up.pt ,    † jlopes@fc.up.pt

## Abstract

Charge transport in disordered two-dimensional (2D) systems showcases a myriad of unique phenomenologies that highlight different aspects of the underlying quantum dynamics. Electrons in such systems undergo a crossover from ballistic propagation to Anderson localization, contingent on the system's effective coherence length. Between the extended and localized phases lies a diffusive crossover in which the charge conductivity is properly defined. The numerical observation of this regime has remained elusive because it requires fully coherent transport to be simulated in systems whose dimensions are sufficiently large to meaningfully split the mean-free path and localization length scales. To address this challenge, we employed a novel linear scaling time-resolved approach that enabled us to derive the dc-transport characteristics and observe the three expected 2D transport regimes — *ballistic*, *diffusive*, and *localized*.

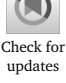

# 1 Introduction

As first understood by P. W. Anderson [1], the existence of quenched disorder can drastically influence how electrons wander along an applied electric field in a metallic system. As it gets stronger, disorder eventually causes all single-particle eigenstates to exponentially localize in space, converting the system into a bulk insulator where charge transport only occurs via the thermally activated long-range hopping of electrons between distant localization centers [2]. While this is a generic behaviour of disordered lattices, the onset of Anderson localization as a function of disorder strength is a process that depends crucially on the system's dimensionality. For one-dimensional (1D) electrons, it is well known that an infinitesimally weak amount of disorder will be strong enough to localize all quantum states [3,4] while, in three-dimensions (3D), a system is generally expected to remain metallic up to a very strong disorder [5–8]. Just between these two situations lie the two-dimensional (2D) systems, which formally define the lower critical dimension for the Anderson localization transition.

In the absence of magnetic fields or spin-dependent scattering [9], it has long been established [10,11] that the sign of loop corrections in the single-parameter scaling equations also leads to an absence of diffusion in disordered 2D lattices. Just as it happens in 1D, all single-particle states are immediately localized by the presence of disorder. However, the effects in charge transport are still qualitatively different in 1D and 2D systems despite their apparent similarities [12–14]. In both cases, there are two different (but interrelated) length scales that control the single-electron dynamics: *i)* the mean-free path ($\ell$) which describes the long-distance decay of the *disorder-averaged single-particle propagator* [15] and *ii)* the wave-function localization length ($\xi$) which is extracted from the long-distance behaviour of the *disorder-averaged two-particle propagator* [16]. In 1D these two scales are basically the same ($\xi \propto \ell$) [12], but in 2D systems they can get exponentially separated in the weak disorder limit ($\xi \propto \ell \exp(\pi k_F \ell/2)$ [15]) driven by angular scattering channels that are absent in 1D. This separation of scales opens up space for a sizeable crossover regime where coherent quantum dynamics happens way below the electronic localization length and, therefore, becomes solely dominated by the much shorter mean-free path (i.e., $\ell < L_\phi < \xi$, where $L_\phi$ is the electron phase coherence length). When this happens, charge transport in a 2D system shows a diffusive character akin to that of a disordered 3D metal below its Anderson transition. Only then, electric transport turns into a local process that makes it possible to define an intensive conductivity - derived from the electronic quantum diffusivity via Einstein's relation - that accurately characterizes the system's response to an applied electric field [17–19]. Despite being a well-established result, a definitive observation of such a diffusive crossover in numerical studies of quantum transport is a challenging problem that has been notoriously absent from the literature. However, this matter must be addressed to guarantee that the electrical conductivity results acquired in simulated 2D systems [20–32] can be appropriately understood from a physical perspective. The aim of this work is to demonstrate that such a crossover can indeed be directly observed in numerical studies of quantum transport in disordered Hamiltonians.

Since the 1980s, quantum transport in solid-state models has been studied by one of two complementary approaches: a mesoscopic approach (the so-called Landauer-Büttiker formulation [33–37]) or a bulk response approach (Kubo's formula and its non-linear generalizations [29, 38–42]). The latter is a very general formulation of quantum transport which, in principle, could be used to accurately describe the electrodynamic properties of arbitrary samples independently of them being coupled to leads or not [21, 43]. Additionally, it can be entirely formulated in terms of lattice Green's functions [39, 40, 42]. Their mathematical convenience sparked the development of very efficient real-space algorithms [25, 44, 45] that are able to numerically compute bulk conductivities of independent electron models with a linear complexity in the number of orbitals. However, despite its practicality, the bulk description of transport has important drawbacks that become especially relevant when attempting to capture non-local mesoscopic effects. For a start, it implicitly assumes there is a well-defined local conductivity in the system, something that does not hold true unless it behaves diffusively. Secondly, when such numerical calculations are performed in finite lattices, an effective linewidth must always be assigned to the system's energy levels to smooth out their discrete spectra [13, 44–48]. This technical detail means that, by design, the method is effectually computing a space average of local conductivities that were determined for different phase coherent regions within the sample [46, 49, 50]. The space averaging of the conductivity has a crucial impact on the results obtained within the non-local mesoscopic regime or deep in the localized regime [49, 51, 52]. Therefore, any result from bulk transport calculations is only physically accurate if the system is in a diffusive transport regime.

In contrast, the mesoscopic approach [33, 34, 53] provides a way to analyse quantum charge transport across a sample as a wave scattering phenomenon without any assumption of locality. In its simplest form, the Landauer-Büttiker approach assumes that the mesoscopic sample is connected between two electrical leads (clean semi-infinite conductors) with the conductance of the entire sample, $G$, being determined by the sample's quantum transmission in the energy band comprised by the electrochemical potentials of the two leads [37]. Being semi-infinite, the leads function as free fermionic baths coupled to the sample which provide the continuous spectrum necessary to eliminate the mean-level spacing. Due to these two factors, the mesoscopic formulation can precisely characterize all the system's transport regimes and makes it possible to distinguish them from the conductance's behavior as the disorder strength or system dimensions are changed. A localized regime, for example, will exhibit log-normal sample-to-sample fluctuations in the conductance, with a typical value that decreases exponentially to zero as the distance between the two leads increases [54, 55]. Regarding the diffusive regime, it is distinguished by a conductance with Gaussian fluctuations and an average value that scales as $G \propto S/L$, where $S$ is the sample's cross-sectional dimension and $L$ is the distance between the leads [56].

Unfortunately, despite its wide scope, the efficiency of mesoscopic transport calculations poses a severe limitation on its application for studying transport in very large systems, a particularly critical point when trying to access 2D diffusive transport regimes. In practice, these calculations are performed through the so-called Caroli-Mier-Wingreen equation [35, 57], which is an exact Green's function representation of the Landauer-Büttiker formula amenable to a real-space formulation that requires the semi-infinite leads to be imposed as non-hermitian boundary conditions on the lattice Hamiltonian (surface-green's functions). These boundary conditions can be computed using iterative decimation [58] or eigenchannel decomposition algorithms [36, 59, 60] (see Lewenkopf and Mucciolo [61] for a review), while the sample's transmission coefficient is obtainable via the real-space Recursive Green's function method (RGFM) [36]. Implementations of the RGFM are generally very memory-efficient algorithms but feature an asymmetric scaling with the system dimensions: typically an $\mathcal{O}(S^3 L)$ time complexity. A more favourable (but more memory expensive) approach is found in the Kwant

transport code [60, 62], which relies on a nested dissection algorithm [63] that shows an $\mathcal{O}\left(S^2 L\right)$ scaling instead. In both cases, the unfavourable non-linear scaling with the cross-sectional dimension is a bottleneck that generally prevents the use of a mesoscopic approach to reach and observe the diffusive crossover regime in simulated 2D samples.

Recently, some proposals [64–66] have been made on how to avoid the limitations of mesoscopic transport approaches by means of altered spectral methods that allow electrical contacts to be embedded on the simulated system [67]. Instead, we take on a different path and tackle the problem by a time-resolved approach in which steady-state transport properties are determined from the stabilized quantum dynamics of the system (coupled to truncated leads) after being biased by an external potential. Time-resolved approaches to quantum transport have seen an increase in popularity, ever since their usefulness for the study of dc-transport was clearly demonstrated in 1D systems [68, 69]. In addition, these techniques can also be used to study more complex electrodynamic effects such as transient current dynamics [70, 71], mesoscopic Bloch oscillations [72, 73] and non-linear optical effects [74–78] (including novel non-linear photo-galvanic effects [79–82]) in simulated tight-binding systems. In this paper, we extend the methodology from [68] to investigate all transport regimes of a wide 2D nanoribbon within a two-contact transport setup. To enable the observability of a diffusive crossover regime, we further refine the time-dependent approach to integrate the effectiveness of stochastic trace evaluation techniques [44, 68] with a bandwidth compression scheme (space-modulation of the hopping) inside the finite leads [83]. Together, these increments allow the accurate simulation of sufficiently wide nanoribbons to demonstrate the 2D diffusive crossover regime.

This paper is structured as follows: Section 2 presents the main components of our toy-model Hamiltonian and provides technical information on the methods used for measuring the charge current and performing the time-evolution of the many-electron system. The bandwidth compression scheme is described in detail in Section 3, initially for 1D systems and then expanded for 2D. The main findings of the paper are presented in Section 4, where the outcomes of our simulations clearly show a 2D diffusive crossover. Lastly, we summarize our findings and offer a future direction for this work in Section 5.

## 2 Details on models and methods

The aim of this work is to study time-dependent quantum transport in a two-dimensional two-terminal system whose geometry we can control. We follow [68] by splitting the system into the sample and finite leads. Even though the leads are finite, their properties are tailored as to accurately reproduce semi-infinite ideal leads. Both the disorder and electric field $E$ exist exclusively inside the sample. The electric field is uniform and is adiabatically engaged from $t = 0$, leading to a potential difference $\Delta V = EL$ across the leads (see Fig. (1)(b)) and giving rise to a current.

### 2.1 Equilibrium Hamiltonian

The underlying Hamiltonian for both the sample and the leads is the two-dimensional square lattice tight-binding Hamiltonian of hopping parameter $w$. The sample has length $L$ and width $S$, and is connected to the left and right leads of length $L_l$ and width $S$ (see Fig. (1)(a)). Before the electric field is turned on, the Hamiltonian can be expressed as a sum of five terms,

$$\mathcal{H}_0 = \mathcal{H}_L + \mathcal{H}_R + \mathcal{H}_S + \mathcal{V}_{LS} + \mathcal{V}_{SR}, \tag{1}$$

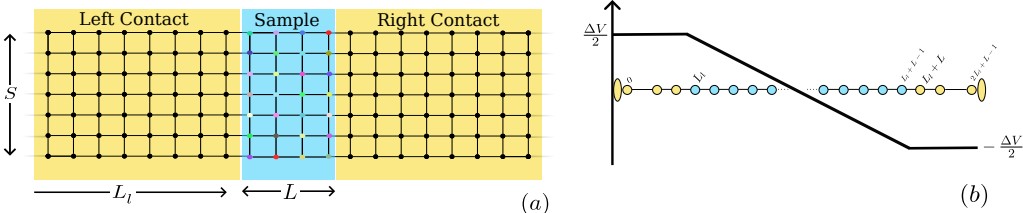

Figure 1: a) Geometry used in our calculations: two-dimensional tight-binding lattice divided into disordered sample and left and right leads, with applied potential $V(x)$. The sample's longitudinal length is noted by L, whereas its width is described by S. Both finite sized contacts possess a lateral length of $L_l$ sites. b) Longitudinal cross-section of the spatial profile of the applied potential.

where $\mathcal{H}_L$ ($\mathcal{H}_R$) stands for the Hamiltonian of the left (right) finite contact, $\mathcal{H}_S$ describes the sample, and $\mathcal{V}_{SL}$ ($\mathcal{V}_{SR}$) describes the boundary hoppings coupling the sample to the left (right) contact. The sample term is written as

$$\mathcal{H}_S = \sum_{x=L_l}^{L_l+L-1} h_x + \sum_{x=L_l}^{L_l+L-2} u_{x,x+1}, \tag{2}$$

where the Hamiltonian of a single slice is described as

$$h_x = \sum_{y=0}^{S-1} \varepsilon(x,y)|x,y\rangle\langle x,y| - w\sum_{y=0}^{S-2}[|x,y+1\rangle\langle x,y| + \text{H.c.}]. \tag{3}$$

The term describing the electronic hopping, $w$, between neighbouring slices is

$$u_{x,x+1} = -w\sum_{y=0}^{S-1}[|x+1,y\rangle\langle x,y| + \text{H.c.}], \tag{4}$$

where $\varepsilon(x,y)$ is an onsite disordered potential whose values are drawn out of a box distribution in the interval $[-W/2, W/2]$ and the lattice parameter was set to unity. In order to mimic the consequences of connecting the central device to semi-infinite leads we employ a spatial modulation of the leads' hoppings and onsite energies (see Section 3 for a detailed explanation). For this reason, we distinguish between vertical ($t_y$) and horizontal ($t_x$) hoppings such that the Hamiltonian of the left lead can be written as

$$\mathcal{H}_L = \sum_{x=0}^{L_l-1} h_{L;x} + \sum_{x=0}^{L_l-2} u_{L;x,x+1}, \tag{5}$$

with

$$h_{L;x} = \sum_{y=0}^{S-1} U(x,y)|x,y\rangle\langle x,y| + \sum_{y=0}^{S-2} t_y(x,y)[|x,y+1\rangle\langle x,y| + \text{H.c.}], \tag{6}$$

and

$$u_{L;x,x+1} = \sum_{y=0}^{S-1} t_x(x,y)[|x+1,y\rangle\langle x,y| + \text{H.c.}]. \tag{7}$$

The right contact is modelled after

$$\mathcal{H}_R = \sum_{x=L+L_l}^{2L_l+L-1} h_{R;x} + \sum_{x=L+L_l}^{2L_l+L-2} u_{R;x,x+1}, \tag{8}$$

with

$$h_{\text{R};x} = \sum_{y=0}^{S-1} U(x,y) |x,y\rangle \langle x,y| + \sum_{y=0}^{S-2} t_y(x,y) [|x,y+1\rangle \langle x,y| + \text{H.c.}], \tag{9}$$

and

$$u_{\text{R};x,x+1} = \sum_{y=0}^{S-1} t_x(x,y) [|x+1,y\rangle \langle x,y| + \text{H.c.}]. \tag{10}$$

Finally, the connection between the three parts of the system is given by the boundary hoppings, which are defined as

$$\mathcal{V}_{\text{SR}} = -w \sum_{y=0}^{S-1} [|L_l + L, y\rangle \langle L_l + L - 1, y| + \text{H.c.}], \tag{11a}$$

$$\mathcal{V}_{LS} = -w \sum_{y=0}^{S-1} [|L_l, y\rangle \langle L_l - 1, y| + \text{H.c.}]. \tag{11b}$$

Before we describe the systematic approach that was taken to simulate the quasi-steady state of the current's time-evolution (see Section 2.2 for details), it is worth noting how the mesoscopic approach referenced in the Introduction would be applied to the study of quantum transport within a sample described by Eq. (2).

**The Landauer-Büttiker method**

For our specific context we consider the two-terminal Landauer formula, which relates the steady-state current traversing from the left to the right lead with the energy integral of the central sample's quantum transmittance, $\mathcal{T}(\varepsilon)$. More precisely, we have

$$I_{\text{L}\to\text{R}}^{Land} = \frac{e}{2\pi\hbar} \int_{-\infty}^{+\infty} d\varepsilon \left[ f_{\text{F}}\left(\varepsilon + \frac{\Delta V}{2}\right) - f_{\text{F}}\left(\varepsilon - \frac{\Delta V}{2}\right) \right] \mathcal{T}(\varepsilon), \tag{12}$$

where $f_{\text{F}}(x) = 1/(1 + \exp((x - \mu)/k_B T))$ is the Fermi-Dirac distribution. Despite the integration over energies being performed from $-\infty$ to $+\infty$ the difference between the Fermi-Dirac functions will severely reduce the effective range of this computation. The integral's range is reduced to a window centered around the Fermi energy, being controlled by the temperature, $T$ and the potential bias, $\Delta V$. From here on forward, the *transmission band* will be defined to be this particular region within the system's spectrum. In the limit $T \to 0K$, it corresponds to a window centered around the Fermi energy, spanning from $-\Delta V/2$ up to $\Delta V/2$. In the absence of interactions, the quantum transmittance of the sample can be expressed within the non-equilibrium transport formalism of Kadanoff-Baym [84] and Keldysh [85], yielding the well-known *Caroli formula* [35]:

$$\mathcal{T}(\varepsilon) = \text{Tr}\left[ \mathbf{G}_\varepsilon^{\text{a}} \cdot \boldsymbol{\Gamma}_\varepsilon^{\text{R}} \cdot \mathbf{G}_\varepsilon^{\text{r}} \cdot \boldsymbol{\Gamma}_\varepsilon^{\text{L}} \right], \tag{13}$$

where $\mathbf{G}_\varepsilon^{\text{a/r}} = \left[ \varepsilon \mp i0^+ - \mathcal{H}_S - \Sigma_\varepsilon^{\text{L}} - \Sigma_\varepsilon^{\text{R}} \right]^{-1}$ is the sample's advanced/retarded single-particle Green's function in the presence of both leads. Each lead dresses the single-particle states of the sample by a self-energy, $\Sigma_\varepsilon^{\text{L/R}}$, whose anti-hermitian part also defines the level-width operators $\boldsymbol{\Gamma}_\varepsilon^{\text{R}}$.

## 2.2 Non-equilibrium time-evolution

To study the dynamics of the system in Fig. (1) (a), we assume that it starts from a thermal equilibrium with temperature, $T$ and common chemical potential, $\mu$. This is described by the single-particle density matrix

$$\rho_0 = \frac{1}{1 + \exp\left[\frac{\mathcal{H}_0 - \mu}{k_B T}\right]}, \tag{14}$$

which is driven out of equilibrium by a potential ramp

$$V(x) = \begin{cases} 1/2, & x < L_l, \\ \frac{1}{L+1}\left(L_l - \frac{1}{2} + \frac{L}{2} - x\right), & L_l \le x \le L_l + L - 1, \\ -1/2, & x > L_l + L - 1, \end{cases} \tag{15}$$

such that $\Delta V \times V(x)$ gives rise to a potential difference of $\Delta V$ between the leads as shown in Fig. (1) (b). The corresponding operator for this potential bias is

$$\mathcal{V} = -e \sum_{x=0}^{L_l+L-1} \sum_{y=1}^{S} V(x) |x,y\rangle \langle x,y|, \tag{16}$$

such that the full time-dependent Hamiltonian is

$$\mathcal{H}(t) = \mathcal{H}_0 + \Delta V f(t) \mathcal{V}, \tag{17}$$

where the temporal profile $f(t)$ encapsulates the full time-dependence of the perturbation. Ultimately, we want to obtain the current traversing the sample, which requires summing up all the contributions of the local currents along a cross-section. Let $\mathcal{I}_{x,y}$ be the operator that represents the electrical current flowing from site $(x, y)$ to $(x+1, y)$:

$$\mathcal{I}_{x,y} = \frac{ew}{i\hbar}\left(|x+1,y\rangle \langle x,y| - |x,y\rangle \langle x+1,y|\right). \tag{18}$$

The expectation value of this operator provides the current $I_{x,y}(t)$ flowing from site $(x, y)$ to $(x+1, y)$ at time $t$ and is expressed as the following trace:

$$I_{x,y}(t) = \mathrm{Tr}\left[\rho(t) \mathcal{I}_{x,y}\right], \tag{19}$$

with $\rho(t)$ being the density matrix evaluated at an arbitrary instant. Since the scope of this paper is to compute a linear response electrical coefficient, from this point forward we will consider a linearized expression for the density matrix. If the perturbation is suddenly turned on at $t = 0$, the resulting current $I_{x,y}^{\mathrm{sud}}$, up to linear order in $\Delta V$ can be computed as

$$I_{x,y}^{\mathrm{sud}}(t) = \int_0^t dt' \dot{I}_{x,y}^{\mathrm{sud}}(t'), \tag{20}$$

where the time derivative of $I_{x,y}^{\mathrm{sud}}$ is defined as

$$\dot{I}_{x,y}^{\mathrm{sud}}(t') = \frac{\Delta V}{i\hbar} \mathrm{Tr}\left[\mathcal{U}_{t'}[\mathcal{V}, \rho_0] \mathcal{U}_{t'}^\dagger \mathcal{I}_{x,y}\right], \tag{21}$$

where $\mathcal{U}_{t'} = e^{-\frac{i}{\hbar}\mathcal{H}_0 t'}$ is the time evolution operator of the unperturbed system from time 0 to $t'$. The trace is evaluated with methods that are based on the Kernel Polynomial Method (KPM) [44, 48, 86] (see Section 2.3). If the profile $f(t)$ is chosen such that $f(0) = 0$ and $f(t \to \infty) = 1$, we will obtain a local current,

$$I_{x,y}(t) = \int_0^t dt' \dot{f}(t-t') I_{x,y}^{\mathrm{sud}}(t'), \tag{22}$$

whose profile is the convolution between a smoothing filter and the time-dependent current obtained with partition-free initial conditions. Owing to the sudden connection of the biasing potential $I^{\text{sud}}_{x,y}(t)$ oscillates along the quasi-steady state plateau. If these oscillations are small compared with the time average of the quasi-steady state, then this quantity agrees perfectly with the Landauer formalism prediction. However, if the system's geometry and disorder strength place it in the localized phase, the average value of the quasi-steady state drops and the amplitude of the oscillations becomes comparable to it. To address this, the uniform electric field is adiabatically connected, with a time dependence given by $f(t)$. The measured transverse current then results from a moving average between $I^{\text{sud}}_{x,y}(t)$, and a kernel, $\dot{f}(t)$. Hereafter, the presented time-dependent results were computed using $f(t) = 1 - e^{-\frac{t}{\tau}}$, where $\tau$ is an adiabatic parameter.

## 2.3 Chebyshev time-evolution method

While Eq. (21) formally tells us how to calculate the current crossing the transverse section of the sample, as a function of time, it still requires the calculation of $\mathcal{U}_t$ and $\rho_0$, which are nontrivial functions of $\mathcal{H}_0$. The trace is replaced by the average over the expectation value of an ensemble of random vectors, and the operators are replaced by a Chebyshev series expansion in powers of the Hamiltonian. Since this series only converges on the open interval between -1 and 1 on the real number line, the Hamiltonian has to be shifted and rescaled such that its eigenvalues lie within this interval. We define $\tilde{\mathcal{H}}_0 = (\mathcal{H}_0 - \lambda)/\Delta$ as the rescaled Hamiltonian, $\lambda$ as the shift which makes the lower and upper bounds of the spectrum symmetric and $\Delta$ as the spectrum range of the Hamiltonian. In practice, $\Delta$ is slightly larger than this to ensure the openness of the interval. In our case, we can set $\lambda = 0$ because the spectrum is symmetrical. With this in mind, the density matrix and the time evolution operators are expanded as

$$\rho_0 = \sum_{m=0}^{\infty} \mu^F_m T_m\left(\tilde{\mathcal{H}}_0\right), \tag{23}$$

$$\mathcal{U}_t = \sum_{m=0}^{\infty} \mu^U_m(t) T_m\left(\tilde{\mathcal{H}}_0\right), \tag{24}$$

where $T_m(x)$ is a Chebyshev polynomial of the first-kind, and the Chebyshev moments, $\mu^F_m$ and $\mu^U_m(t)$, are defined as

$$\mu^F_m = \frac{2}{1+\delta_{m,0}} \int_{-1}^{1} dx \frac{T_m(x)}{\pi\sqrt{1-x^2}\left[1+\exp\left(\frac{x-\mu/\Delta}{k_B T/\Delta}\right)\right]}, \tag{25}$$

$$\mu^U_m(t) = \frac{2}{1+\delta_{m,0}} (-i)^m J_m(w\Delta t/\hbar), \tag{26}$$

where $J_m(x)$ is a Bessel function of the first kind. The coefficients of the Fermi function contain the full information about the temperature and chemical potential of the initial thermal state. In a similar way, the time dependence of the time evolution operator is entirely captured in the $\mu^U_m(t)$ coefficients.

The trace in Eq. (21) when evaluated in real-space can be reduced to the computation of the following matrix elements due to the fact that the current operator is local:

$$\dot{I}^{\text{sud}}_{x,y}(t) = -2\Delta V \frac{ew}{\hbar^2} \operatorname{Re}\left[\langle x, y|\mathcal{U}_t \mathcal{V}\mathcal{U}_t^\dagger \rho_0|x+1, y\rangle - \langle x+1, y|\mathcal{U}_t \mathcal{V}\mathcal{U}_t^\dagger \rho_0|x, y\rangle\right]. \tag{27}$$

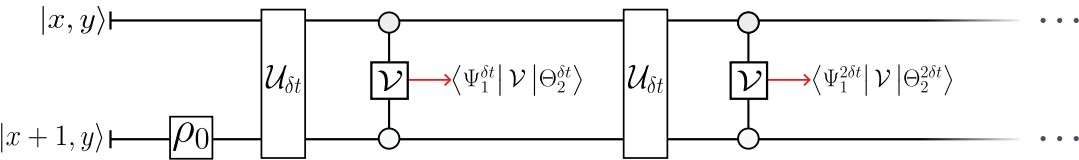

Figure 2: Schematics of the operations required to compute the first term of a local current's time-derivative.

Each of these terms can be computed by using the series expansion of $\rho_0$ and $\mathcal{U}_t$ and the recursive properties of the Chebyshev polynomials. More precisely, given an arbitrary state $|\psi\rangle$, we define $|\psi_m\rangle$ from the action of the Chebyshev operator $T_m$ on $|\psi\rangle$ and we use the recursive properties of these polynomials to relate the different $|\psi_m\rangle$ among themselves, as such:

$$|\psi_m\rangle \equiv T_m\left(\tilde{\mathcal{H}}_0\right)|\psi\rangle = 2\tilde{\mathcal{H}}_0|\psi_{m-1}\rangle - |\psi_{m-2}\rangle, \quad \text{with} \quad |\psi_0\rangle = |\psi\rangle, \text{ and } |\psi_1\rangle = \tilde{\mathcal{H}}_0|\psi\rangle.$$

Therefore, the truncated approximations of the action of $\rho_0$ and $\mathcal{U}_t$ on $|\psi\rangle$ simply read

$$\rho_0|\psi\rangle = \sum_{m=0}^{M_f-1} \mu_m^F|\psi_m\rangle, \text{ and } \mathcal{U}_t|\psi\rangle = \sum_{m=0}^{M_t-1} \mu_m^U(t)|\psi_m\rangle, \tag{28}$$

where $M_f$ ($M_t$) is the truncation order for the expansion of the $\rho_0$ ($\mathcal{U}_t$) operator. The calculation of $I_{x,y}(t)$ requires the evaluation of $\dot{I}_{x,y}^{\text{sud}}(t)$ along a discrete set of times $t_n = n\delta t$ in the integration interval from 0 to $t$. This process would require computing the time evolution operator for all times $t_n$. To avoid this, we instead compute $\dot{I}_{x,y}^{\text{sud}}(t_n)$ in increments of $\delta t$ reusing the objects that had already been used for $\dot{I}_{x,y}^{\text{sud}}(t_{n-1})$. This procedure can be summarized as follows (see Fig. (2) for a diagrammatic representation):

1. Define the four Chebyshev vectors:

$$\left|\Psi_1^0\right\rangle \equiv |x,y\rangle, \quad \left|\Psi_2^0\right\rangle \equiv |x+1,y\rangle, \quad \left|\Theta_1^0\right\rangle \equiv \rho_0|x,y\rangle, \quad \left|\Theta_2^0\right\rangle \equiv \rho_0|x+1,y\rangle. \tag{29}$$

2. Time evolve all four vectors in a series of time-steps (sized $\delta t$):

$$\begin{aligned} \left|\Psi_1^{t_{n+1}}\right\rangle &= \mathcal{U}_{\delta t}\left|\Psi_1^{t_n}\right\rangle, \\ \left|\Psi_2^{t_{n+1}}\right\rangle &= \mathcal{U}_{\delta t}\left|\Psi_2^{t_n}\right\rangle, \\ \left|\Theta_1^{t_{n+1}}\right\rangle &= \mathcal{U}_{\delta t}\left|\Theta_1^{t_n}\right\rangle, \\ \left|\Theta_2^{t_{n+1}}\right\rangle &= \mathcal{U}_{\delta t}\left|\Theta_2^{t_n}\right\rangle. \end{aligned} \tag{30}$$

3. At each time-step evaluate the time-derivative of the current as follows:

$$\dot{I}_{x,y}^{\text{sud}}(t_n) = -2\Delta V \frac{ew}{\hbar^2} \text{Re}\left[\left\langle\Psi_1^{t_n}\right|\mathcal{V}\left|\Theta_2^{t_n}\right\rangle - \left\langle\Psi_2^{t_n}\right|\mathcal{V}\left|\Theta_1^{t_n}\right\rangle\right]. \tag{31}$$

4. Repeat the procedure until time $t$ has been reached, and at the end numerically integrate $\dot{I}_{x,y}^{\text{sud}}(t_n)$. The result will be the linear current from site $(x,y) \to (x+1,y)$.

### 2.3.1 Reducing numerical complexity

If the unperturbed Hamiltonian is represented as a sparse matrix in some basis (usually in real-space), the numerical calculation of either expression in Eq. (28) involves only sparse matrix-vector operations and, thereby, has an $\mathcal{O}(MN)$ time complexity, where $N$ is the dimension of the Hamiltonian and $M$ is the truncation order for that operator. Note that the numerical evaluation of $\dot{I}^{\mathrm{sud}}_{x,y}(t)$ requires a number of operations proportional to $N{\times}(TM_t+M_f)$, where $T$ is the number of points used in the time discretization. For fixed $T$, $M_f$ and $M_t$, the time complexity is $\mathcal{O}(N)$. The complete transverse current current requires $S$ independent calculations, one for each $y$, totalling a time complexity of $\mathcal{O}(NS)$ and making it unfavourable for wide systems with large $S$. This complexity can be brought down to $\mathcal{O}(N)$ with the use of random numbers, in a similar fashion to KPM. The linear combination of position operator eigenstates with un-correlated random coefficients (noted as $|\xi_x\rangle$) with variance one ($\overline{\xi_y \xi_{y'}} = \delta_{yy'}$), Eq. (32), is a core component for the stochastic measurement of the trace in Eq. (19)

$$|\xi_x\rangle \equiv \sum_{y=1}^{S} \xi_y |x,y\rangle \,. \tag{32}$$

Furthermore, the definition of a translation operator by a unit cell along the longitudinal direction ($\mathcal{T}$), which acts on single-particle position eigenstates as

$$\mathcal{T}|x,y\rangle = |x+1,y\rangle \,, \tag{33}$$

enables this stochastic procedure to be expressed as

$$I_x(t) \equiv \sum_{y=1}^{S} I_{x,y}(t) = \frac{2ew}{\hbar} \mathrm{Im}\, \overline{\langle \xi_x | \rho(t) \mathcal{T} |\xi_x\rangle}. \tag{34}$$

Since $\langle \xi_x | (\dots) | \xi_x \rangle$ is a random number, it has a variance associated with its distribution. To get a sufficiently small error bar, an average across a large number $R$ of random vectors $|\xi_x\rangle$ needs to be performed. For this procedure to be more advantageous relative to summing $I_{x,y}$ over $y$, $R$ has to be smaller than $S$ [87].

## 3 Bandwidth compression schemes

As it was shown in [68], the finite dimension of the Hilbert space imposes constraints on the time-dependent simulations with finite-sized leads. In particular, the duration of the quasi-steady-state plateau ($T_r$) is limited by the reflection time of the Fermi level states at the lead's terminations, $T_r = 2\hbar L_l / \sqrt{w^2 - \varepsilon_F^2}$, with $\varepsilon_F$ being the system's Fermi energy. This is a problematic feature of time-dependent simulations with finite-sized leads since $T_r$ may prove to be too short for transients to die out, making it impossible for the current to converge to the Landauer quasi-steady state. An intuitive solution is to increase the size of the finite leads, and consequentially reduce the mean-level spacing within the transmission band, increasing $T_r$. However, due to the number of operations being proportional to the number of Hilbert space elements, whenever we increase the size of the leads we will also raise the required computational effort. Therefore, we introduce an alternative procedure that decreases the mean-level spacing within the transmission band and, after the initial transient dies out, mimics a de facto steady-state that would be reached on the limit of semi-infinite leads. We will discuss how this comes about in the following sections, building up the intuition from the 1D case.

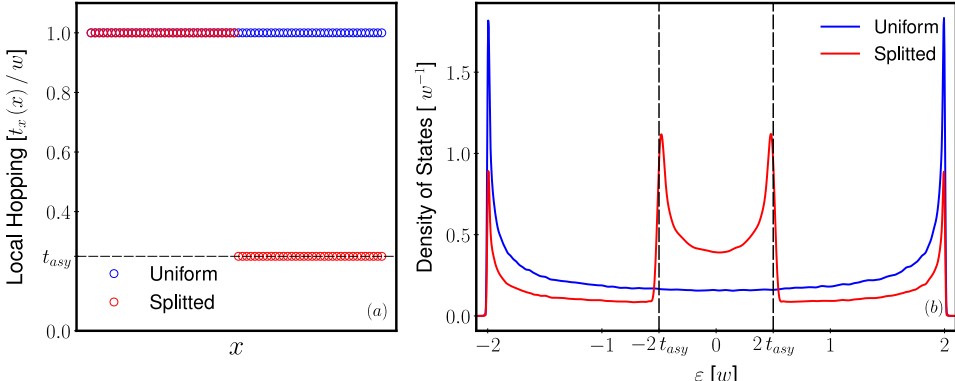

Figure 3: In $(a)$ we represent the hopping term for two different 1D tight-binding chains. Whereas the blue plot corresponds to an unchanged system, on the red curve we are altering the value of the hopping term at half the sites of the chain. In $(b)$ we compute the density of states (DoS) for these two models. On one hand, we retain the DoS for the unchanged system (with half of the total number of states), while on the other hand we have the density of states of a system whose hopping term is $t_{asy}$. This result shows that we are capable of increasing the DoS on a region within the energy spectrum that is solely controlled by $t_{asy}$.

## 3.1 Reflection time enhancement in 1D systems

As previously stated the main purpose of the introduction of a bandwidth compression scheme is to reduce the mean-level spacing within the transmission band. In order to motivate this procedure, we can look at a one-dimensional toy model. Let us consider, as depicted in Fig. 3 $(a)$, a tight-binding chain where the first half of the system is characterized by a constant hopping, $t$ and the second half by $t_{\text{asy}}$. If one computes the density of states (DoS) for this partitioned system we will see that, although we maintain the signatures of the original system, with local maxima at $-2w$ and $2w$, we are adding to this DoS, the DoS of the tight-binding chain with hopping $t_{asy}$. Consequently, this change of the hopping term leads to an increase of the DoS in a region of the system's spectrum that is limited by $-2t_{asy}$ and $2t_{asy}$. Therefore, it is anticipated that a heuristic construction of $t_{asy}$ allows for a reduction of the transmission band's mean-level spacing. Complementary to this parameter, we introduce a sewing function that should be able to glue the two different asymptotic regimes that we are looking for. On the one hand, it has to retain the hoppings set to unity inside the sample and on another hand, deep within the leads' profile it should equate to $t_{asy}$ (ensuring the dense population of the transmission band). A possible candidate for such parametrization is

$$f(x; s; \sigma) = \frac{1}{2}\left(\text{erf}\left[\frac{s + \sigma - x}{\sigma}\right] - \text{erf}\left[-\frac{s + \sigma + x}{\sigma}\right]\right),\tag{35}$$

where $\{s, \sigma\}$ are two free parameters that may be tuned to optimize the results.

The construction of the spatial modulation scheme can be attained by directly constraining the upper limit of the local half-bandwidth, $V_T$, as

$$V_T \equiv \min\left(\varepsilon_F + 2t_{asy}, 2w\right),\tag{36}$$

where the minimum with $2w$ is done in order to prevent the appearance of states outside the original bandwidth. The lower bound for this parametrization is similarly obtained from

$$V_B \equiv \max\left(\varepsilon_F - 2t_{asy}, -2w\right),\tag{37}$$

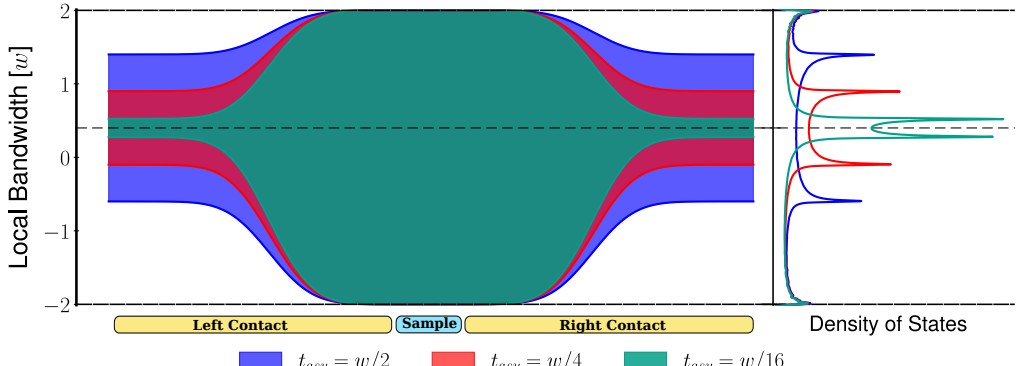

Figure 4: Schematics of the bandwidth compression introduced by the hopping's modulation within a one dimensional system. The density of states increases in the neighbourhood of the Fermi energy, and the separation between the its maxima is approximately $4t_{asy}$, where $t_{asy}$ corresponds to the chosen asymptotic value of the hopping term.

such that, at each point, the difference between $V_T$ and $V_B$ is the system's local bandwidth. The spatial dependency of the hopping term was chosen to be

$$t_x(x) = \frac{V_T(x) - V_B(x)}{4}, \tag{38}$$

where

$$
\begin{aligned}
V_T(x) &= V_T + (2w - V_T) f(x; s; \sigma), \\
V_B(x) &= V_B - (2w + V_B) f(x; s; \sigma).
\end{aligned}
\tag{39}
$$

The sole application of Eq. (38) will only increase the density of states close to zero energy. Consequently, as one has to decrease the mean-level spacing around the system's Fermi energy, the local potential term should also possess a spatial modulation, that ensures the correct population of the transmission band. Employing the spatial dependencies shown in Eq. (39) and considering that the lead's modulated profile should not introduce states whose energy is not comprised within the original system's bandwidth, one can build a modulation of the local onsite energies as

$$U(x) = \frac{V_T(x) + V_B(x)}{2}, \tag{40}$$

such that Eq. (38) and Eq. (40) will progressively compress the total bandwidth around the Fermi energy, as one observes in Fig. 4. The application of this bandwidth compression scheme is responsible for increasing the time-dependent current's reflection time, as it is shown in Fig. 5, where we have represented the current's time-evolution for the same mesoscopic setup but we consider modulation profiles whose asymptotic values were gradually smaller. According to Fig. 5(a), choosing smaller values of $t_{asy}$ systematically increases the reflection time, without affecting the value of the quasi-steady-state current. However, this improvement comes to a halt once $t_{asy}$ is so small that the asymptotic bandwidth of the modulated leads becomes smaller than the transmission band.[1] At this point, the quasi-steady state of transport begins to decay, as it is observed in Fig. 5(b).

---

[1]Due to the temperature dependence of the Fermi-Dirac distributions a suitable choice for the hopping's asymptotic value is $t_{asy} = 11 k_B T$.

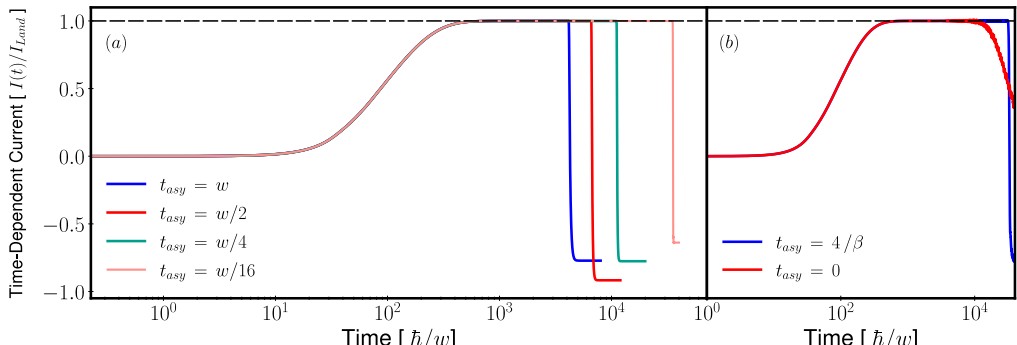

Figure 5: Representation of the current time-evolution for the 1D limiting case. The simulations were performed with $\varepsilon_F = -0.2w, L = 128, \tau = 100, \beta = 1024 w^{-1}$ and $W = 0.3w$ for $(a)$ $L_l = 4096$ and $(b)$ $L_l = 512$. It is shown that a decrease in the asymptotic value of the hopping enables the extension of the current's plateau. Whenever the chosen value of $t_{asy}$ is too small, we are no longer able to ensure that all the states covering the transmission band contribute to the maintenance of the quasi-steady state. Consequently, in $(b)$ an earlier reflection is observed when $t_{asy} = 0$. The normalization, $I_{Land}$, was computed using the RGF method.

## 3.2 Extension to 2D systems

Analogous to subsection 3.1 smooth boundary conditions in 2D systems must compress the density of states of the system within the transmission band. If one introduces the bandwidth compression scheme leaving $t_x(x)$ and $U(x)$ defined by Eq. (38) and Eq. (40), while keeping the vertical hoppings, $t_y$, unchanged and set to $w$ we would not be able to resolve the current's quasi-steady state. This statement is verified in the blue and green plots of Fig. 6(a). Complementary to this information, in Fig. 6(b) we show that we are not able to increase the density of states on the transmission band. Instead each local maxima is centered around $\varepsilon_F - 2w \cos k_y$, where $k_y$ are the allowed wave-numbers of a hard-wall tight-binding chain. In order to understand why this is happening we may look at the Hamiltonian of a clean system. Since the spatial dependence of the longitudinal hoppings is independent of the $y$ coordinate, we can factorize this Hamiltonian as

$$\mathcal{H} = \sum_{k_y} \mathcal{H}^{1D}(k_y) \otimes |k_y\rangle\langle k_y|, \tag{41}$$

where

$$\mathcal{H}^{1D}(k_y) = \sum_x t_x(x)|x+1\rangle\langle x| + \text{H.c.} + \sum_x \left[U(x) - 2w \cos k_y\right]|x\rangle\langle x|. \tag{42}$$

Therefore, the introduction of smooth boundary conditions along the longitudinal hoppings alone, will compress each strip of the nanoribbon in an independent manner, controlled by the diagonal term in Eq. (42). The motivation towards the solution that we have implemented is best realized by restricting the possible functional forms of the vertical hoppings, $t_y(x, y)$ to be functions of $x$ alone, $t_y(x, y) \rightarrow t_y(x)$, while maintaining $t_x(x)$ and $U(x)$ defined by Eq. (38) and Eq. (40). Additionally, we considered that $t_y(x)$ should be constructed from Eq. (38), whose asymptotic value, $t_{y,asy}$, should be set as a small constant.[2] The red plot in Fig. 6(a) and (b) confirms that this prescription is able to extract the Landauer conductance from a

---

[2] We chose to fix it to $t_{asy}/16$. Setting $t_{y,asy} = 0$ would imply that $k_y$ stopped being a good quantum number at the beginning of a lead, since the Fermi surface for an eigenstate of energy, $\varepsilon$, would be solely defined by $k_x$, $2t_{asy} \cos k_x = \varepsilon$.

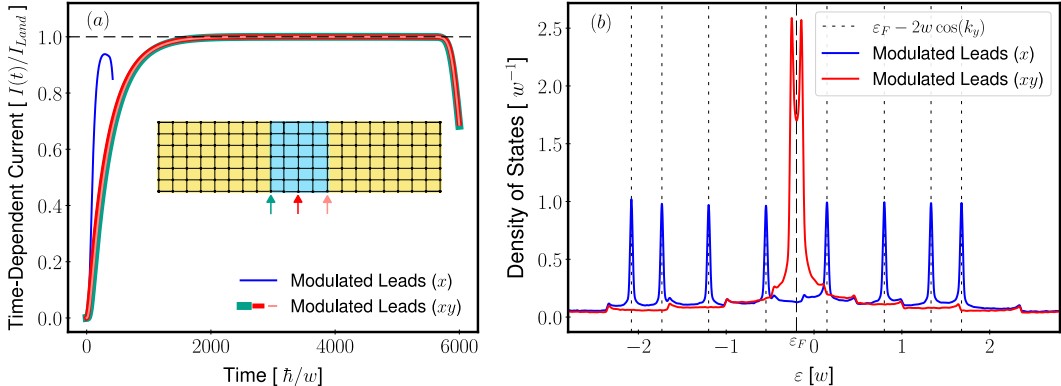

Figure 6: $(a)$ Time-dependent transverse current on a disordered nanoribbon for different simulation setups: bare leads, modulated on the $xx$ direction and modulated on both directions. In $(b)$ it is shown that only when $t_y$ is very small - deep within the leads' profile - we are able to compress the energy levels close to the Fermi energy. The simulations were performed with $\varepsilon_F = -0.2w$, $S = 8$, $L = 128$, $\Delta V = 10^{-6}w$ and $W = 0.2w$ for $L_l = 512$. Akin to Fig. 5, $I_{Land}$, was evaluated using the RGF method.

time-resolved approach in 2D systems and increase the density of states in the transmission band.

To justify the physical grounds for this heuristic solution we begin by looking at an exact eigenstate of the modulated lead, $|\psi_\alpha\rangle$, with energy $\varepsilon_\alpha$. The projection of such state onto a quantum state for which $k_y$ is a good quantum number, $\left|\lambda_{k_y}\right\rangle$, will be a sinusoidal function

$$\psi_{\alpha,k_y}(x) \equiv \left\langle \lambda_{k_y}|\psi_\alpha\right\rangle = A\sin(k_x x + \phi), \tag{43}$$

where $A$ is the wavefunction's amplitude, $\phi$ is its phase and $k_x$ is the allowed wave-number along the longitudinal direction. The bandwidth compression scheme for 2D systems can be seen as a way to have all (or most) eigenstates of the system with finite leads concentrated in energies that cover the transmission band, while having $\{k_x, k_y\}$ locally well defined near the sample, where the hopping is constant. Therefore it is expected that the projected eigenstate to have a well-defined $k_x$ in that region and its value is such that it occupies the Fermi surface. Although the presence of modulation does not permit the definition of $k_x$ as a good quantum number, we still aim at defining a local value for it using the following procedure.

The propagation of a projected eigenstate from the beginning of the left lead up to the sample can be expressed through the transfer matrix, $\mathbb{T}_x^{k_y}$, as

$$\begin{bmatrix} \psi_{x+1} \\ \psi_x \end{bmatrix} = \mathbb{T}_x^{k_y} \begin{bmatrix} \psi_x \\ \psi_{x-1} \end{bmatrix}, \tag{44}$$

with

$$\mathbb{T}_x^{k_y} = \begin{bmatrix} \frac{\varepsilon_F - 2t_y(x)\cos k_y - U(x)}{t_x(x)} & -\frac{t_x(x-1)}{t_x(x)} \\ 1 & 0 \end{bmatrix}. \tag{45}$$

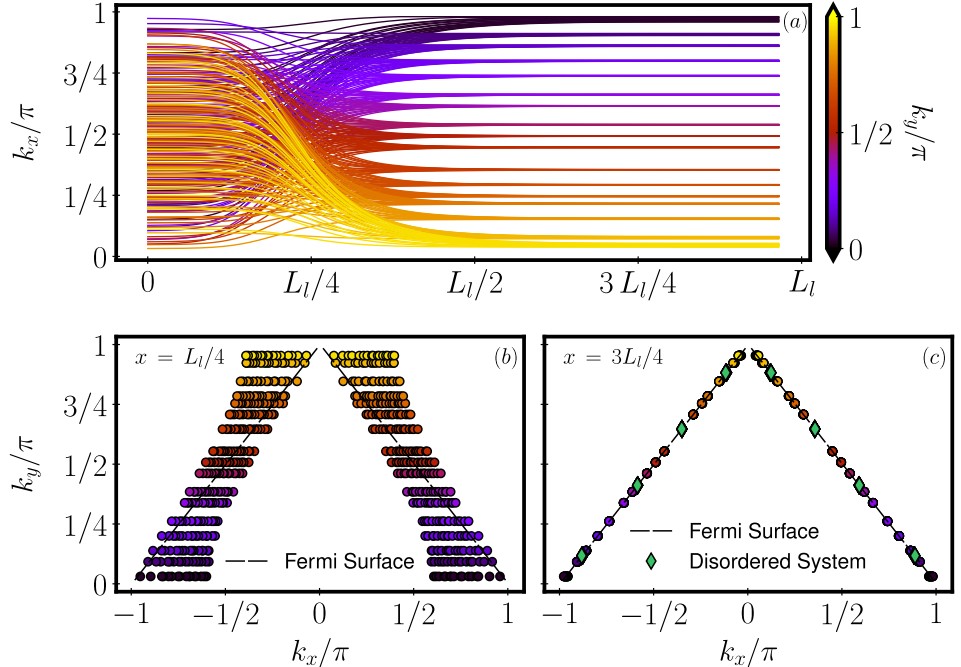

Figure 7: $(a)$ Representation of $k_x(x)$ for a set of fixed transverse momenta, $k_y$. In $(b)$ and $(c)$ we observe cross-sectional cuts of $(a)$ at $x = L_l/4$ and $x = 3L_l/4$. The green data sets shown in $(c)$ verify that this approach is also valid for the case in which the leads are connected to a disordered sample. All panels were computed with $L_L = 512$, whereas the disordered data was evaluated with $S = 16$ and $W = 1.5w$.

Therefore, knowing the projected wave-function in two adjacent sites, $x$ and $x + 1$, we could extend the projected eigenstate, by the multiplication of the transfer matrix associated to each lattice site. In fact, if one wishes to obtain the wave-function, $M$ sites to the front, it is equivalent to the repeated application of the transfer matrix

$$\begin{bmatrix} \psi_{x+M} \\ \psi_{x-1+M} \end{bmatrix} = \prod_{i=0}^{M-1} \mathbb{T}_{x+i}^{k_y} \begin{bmatrix} \psi_x \\ \psi_{x-1} \end{bmatrix}. \tag{46}$$

To define a local $k_x(x)$ we construct an extended state from the successive multiplication by the matrices, $\mathbb{T}_x$,

$$\begin{bmatrix} \psi_{x+M} \\ \psi_{x-1+M} \end{bmatrix} = \left[ \mathbb{T}_x^{k_y} \right]^M \begin{bmatrix} \psi_x \\ \psi_{x-1} \end{bmatrix}. \tag{47}$$

Since these matrices have constant hoppings they generate semi-infinite states that are harmonic functions and have a well-defined $k_x$. This will be our definition of $k_x$: it is the one generated by the local transfer matrix, $\mathbb{T}_x^{k_y}$.

In Fig. 7$(a)$ we show the spatial dependency of $k_x$, as one constructs and fits these extended states at each lattice site. The different realizations of the colour palette marks a choice of $k_y$ and as we have previously stated, within the regions in which the hoppings and onsite potential are severely compressed, we do not have a correlation between $k_x$ and $k_y$. This characteristic is evident by the lack of coherence between the distribution of colours for the regions of the plot that are to the left of $L_l/4$. The progression along the profile of the lead brings about a smooth change of $k_x$ that terminates on a constant value. This is precisely the one permitted by the condition that each pair, $\{k_x, k_y\}$, belongs to the Fermi surface of the infinite lead, as one gets closer to the central device. This conclusion is also pinpointed at Fig. 7$(b)$ and $(c)$

where we see cross-sectional cuts of panel $(a)$. In $(b)$ we show that the pairs $\{k_x, k_y\}$ do not lie within the Fermi surface. Contrastingly, in $(c)$ we are sufficiently close to the sample and each value of $k_x$ that was previously scattered along the Brillouin zone is now correctly mapped towards the Fermi surface.

This was not only verified for a modulated lead but also for the full system with modulated leads connected to a disordered sample. Taking the eigenstates within the transmission band and projecting them on $k_y$, we get $k_x$ values that are on the Fermi surface, as seen in the green diamonds of Fig. 7$(c)$.

## 4  Emergence of a diffusive transport regime

Using the setup described earlier, we analyse the conductance, $G(L, S)$ of disordered systems as a function of their longitudinal length, $L$, and width, $S$, at a fixed Anderson disorder strength, $W = 0.9w$. The time-resolved approach described in Section 3 is crucial to measure the conductance for larger systems, $S \geq 2048$, whereas an implementation of the RGF method was employed for systems with smaller cross-sections. In the diffusive regime the conductance should scale as $G = \sigma S/L$, where $\sigma$ is the system's conductivity. Therefore, by studying the scaled conductance, $g \equiv GL/S$, one can unmistakably identify the system's diffusive behaviour: it occurs when $g$, as a function of $L$ for fixed $S$, remains constant, matching the conductivity.

Each curve in Fig. 8 shows the rescaled conductance's scaling behaviour with the device length, for fixed $S$. Additionally, each panel features vertical dashed lines indicating estimations of both the mean-free path, $\ell$, and localization length, $\xi$. The latter is only displayed in $(a)$ because, only for this particular set of parameters, $\xi$ falls within the range of the simulated device lengths. The mean-free path was estimated by studying the disorder-averaged Green's function, $G^{\mathrm{r}}$, of a large periodic disorder sample, with identical hoppings and disorder strength to Eq. (2). This quantity was obtained from the relation between the disordered-averaged Green's function and its clean counterpart [88], $G^{\mathrm{r}}_{\mathrm{cl}}$,

$$\left| \left\langle G^{\mathrm{r}}\left(\mathbf{x}, \mathbf{x}', \varepsilon_F\right) \right\rangle \right| = \left| G^{\mathrm{r}}_{\mathrm{cl}}\left(\mathbf{x}, \mathbf{x}', \varepsilon_F\right) \right| e^{-\frac{|\mathbf{x} - \mathbf{x}'|}{2\ell}}, \tag{48}$$

where $\langle \dots \rangle$ represents disorder averaging and $G^{\mathrm{r}}\left(\mathbf{x}, \mathbf{x}', \varepsilon_F\right) = \langle \mathbf{x} | G^{\mathrm{r}}(\varepsilon_F) | \mathbf{x}' \rangle$. We have approximated the measurement of the mean-free path in the thermodynamic limit by employing twisted boundary conditions in both directions. By doing so, $G^{\mathrm{r}}$ was averaged over different $512 \times 512$ disordered supercells, while $G^{\mathrm{r}}_{\mathrm{cl}}$ was solely averaged over different twist-angle configurations. Thereafter, the mean-free path was computed by fitting the exponential decay shown in Eq. (48). Whenever charge transport is compatible with the ballistic regime $\langle g \rangle$ becomes exponential in $\log L$. This dependency is observed throughout the plots in Fig. 8, for $L < \ell$. The fitted values of $\ell$ mark the end of the ballistic behaviour and for $L > \ell$ the ballistic-diffusive crossover begins.

The estimation of the localization length can be directly made from the conductance scaling behaviour, since for a localized system, $G \sim S \exp\left(-2L/\xi\right)$. The vertical line shown in Fig. 8$(a)$ was extracted from the normalized conductance curve with the highest cross-section, $S = 1024$. It is shown that due to the closeness between the mean-free path and localization length scales we are not able to observe the formation of a diffusive plateau. From this panel to $(b)$ we move from $\varepsilon_F = -3.985w$ to $\varepsilon_F = -3.8w$, which increases the distance between $\ell$ and $\xi$. Despite this, the separation between these scales is still not sufficient for a sizeable diffusive regime to be observed. We further note that the plot with $S = 8192$ doesn't span all values of $L$. Currently the simulation of non-equilibrium currents on localized samples is still at an early stage of development. We have observed that the initial transient associated with the non-equilibrium current significantly increases as one moves into the localized regime. This particular challenge will be the focus of future publications.

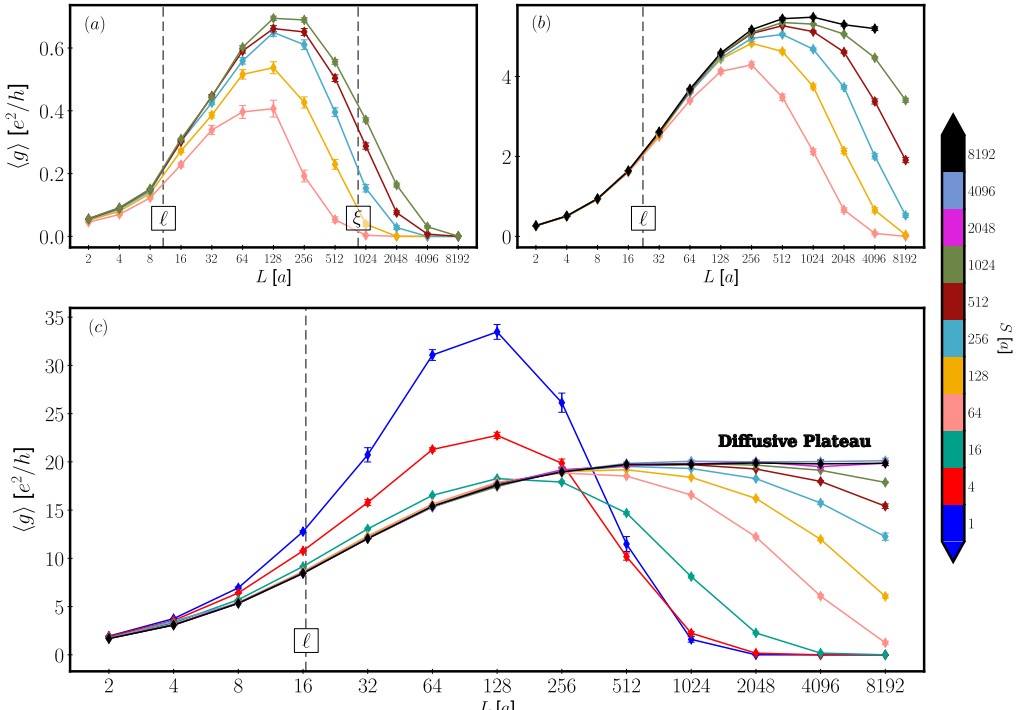

Figure 8: Disordered-averaged scaled conductance as a function of the mesoscopic devices geometries, for a fixed Anderson disorder strength, $W = 0.9w$. In $(a)$ we have fixed $\varepsilon_F = -3.985w$, whereas in $(b)$ we have $\varepsilon_F = -3.8w$ and in $(c)$ $\varepsilon_F = 0.0w$. Only at half-filling, the mean-free path and localization length scales are sufficiently far apart, so that we are able to observe the formation of a diffusive plateau.

The localization length was drastically increased in Fig. 8$(c)$ where we moved to the half-filling case, $\varepsilon_F = 0.0w$. For small $S$, we find quasi one-dimensional behaviour. As $L$ is increased, the conductance decreases slower than $1/L$, causing an initial increase in the disordered-averaged normalized conductance, $\langle g \rangle$. For sufficiently large $L$, a localized regime takes over and $\langle g \rangle$ drops to zero. In quasi one-dimensional systems where $S < \ell < \xi$, it is commonly accepted that the proximity between the mean-free path and localization length scales hinders the development of the diffusive regime. As S increases, the mean free path grows; simultaneously, the localization length also increases at a much faster pace due to the exponential dependence. This separation manifests itself in the broadening of the maximum of $\langle g \rangle$. For sufficiently large $S$ ($S \gtrsim 512$), several things happen:

1. $\xi$ becomes large enough that the localized regime is unobservable within the range of parameters we were able to simulate.

2. As $\ell$ converges to the two-dimensional value with increasing $S$, all the curves collapse into the same curve.

3. A diffusive regime develops for $L \gtrsim 2048$, where $\langle g \rangle$ is constant and equal to the conductivity $\sigma$.

Another common method to determine the zero temperature longitudinal conductivity of a sample is through perturbation theory using the Kubo-Greenwood formula [13,89] in a large periodic and disordered system without leads:

$$\sigma_{xx}(E) = \frac{e^2 \pi \hbar}{\Omega} \text{Tr}[\delta(E - \hat{H})\hat{V}_x \delta(E - \hat{H})\hat{V}_x], \tag{49}$$

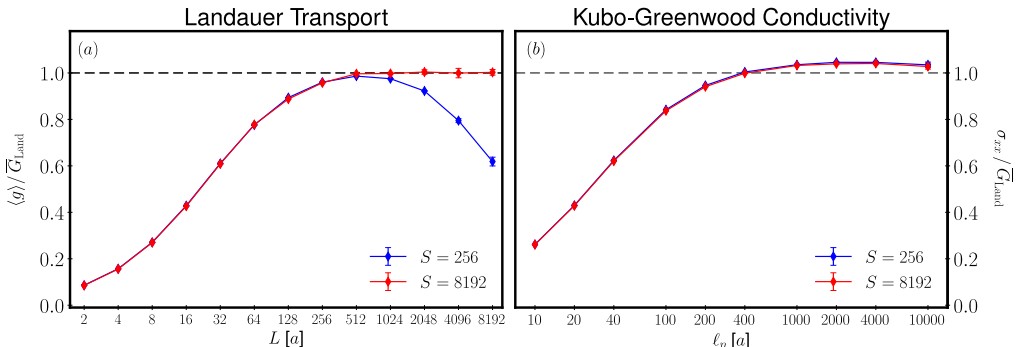

Figure 9: Direct comparison between the scaling behaviour, at half-filling, between the conductance $(a)$ and the Kubo-Greenwood conductivity $(b)$. The plots are normalized by the extracted Landauer conductivity, $\overline{G}_{\text{Land}}$.

where $\hat{V}_x$ is the velocity operator in the longitudinal direction. To establish a close connection to our time-dependent results, we chose a system with exactly the same geometry, hopping and disorder parameters as the sample in Eq. (2). The transverse direction has open boundary conditions, whereas the longitudinal direction has twisted boundary conditions.

The numerical method described in this manuscript is based on quantum transmission and consistently describes both local and nonlocal transport regimes. In contrast, Eq. (49) assumes from the start that transport is local, i.e. diffusive, where a bulk conductivity is properly defined. The numerical evaluation of Eq. (49) for finite systems relies on introducing a phenomenological inelastic parameter, $\eta$, that corresponds to the resolution of a single Dirac-Delta. The results obtained within this framework are highly dependent on the value acquired by $\eta$ [21], and the common procedure is to study the conductivity as a function of this parameter [13]. To obtain physical results, $\eta$ should be larger than the finite system's mean-level spacing, and because this formulation scales linearly with the geometry of the system [86, 90] it is a very powerful tool from a numerical standpoint.

In Fig. 9$(a)$ we reprised two plots from Fig. 8, while in $(b)$ we represented the Kubo-Greenwood conductivity's scaling with the spectral resolution, $\eta$, computed with KITE [48]. When this quantity is determined using Eq. (49), we do not have access to a concrete length scale. It is common to define an effective inelastic length scale, $\ell_\eta$, as $\ell_\eta = \hbar v_F / \eta$, where $v_F$ is the Fermi velocity. Despite the underpinning assumptions of the Kubo approach to quantum transport, Eq. (49) is broadly expected to yield the same results as any transmission-based approach whenever the system's response to the applied electric fields becomes local or, equivalently, if charge transport happens diffusively across the system. This comparison is made explicit in Fig. 9, where it is shown that, while the $\sigma_{xx}$ Kubo conductivity initially grows with $\ell_\eta$ (resembling a ballistic transport regime), it eventually settles into a plateau for intermediate values of this scale suggesting a diffusive behaviour with a size-independent conductivity. Even though the plateau of the Kubo conductivity bears striking similarities to the diffusive plateaus obtained earlier for the Landauer conductance of the two-terminal devices, it is worth remarking that the two values do not perfectly agree, showing an approximately 4% relative mismatch in this particular setup. This minute difference can be attributed to the fact that the diffusive behaviour in two-dimensional systems only exists as a crossover regime, as any disordered two-dimensional electron gas is ultimately localized in the absence of external magnetic fields [10]. Hence, the assumption of local transport in a two-dimensional nanoribbon is only approximately true.

As a final remark, the comparison in Fig. 9 further highlights the very different way in which localization affects the transport results within a Kubo or Landauer approach. The plots present numerical results obtained for nanorribons having two different cross-sections, $S = 256$ and $S = 8192$, corresponding to localization lengths that differ by orders of magnitude. In the two-terminal setup, Fig. 9(a), we see a clear difference in the rescaled conductance for $L > 512$, revealing the two quantum transport regimes: the localized ($S = 256$) and the diffusive ($S = 8192$). From the Kubo-Greenwood results, Fig. 9(b), both geometries present an undistinguishable behaviour, exposing the difficulties of the Kubo-Greenwood formalism to capture localization effects in mesoscopic systems.

## 5 Conclusion and outlook

In this paper, we obtained the conductance of a disordered sample connected to leads, in a two-dimensional tight-binding square lattice, as a function of its geometry. The methods developed in this work enabled us to accurately obtain the conductance of samples with cross-sections ranging from one to several thousand unit cells, allowing a clear separation of scales between the localization length and the mean free path. Thus, a diffusive regime is observable across a wide range of geometries and a two-dimensional conductivity can be defined. Our method shows perfect agreement with the Landauer formula and very good agreement with the Kubo-Greenwood formula in the diffusive regime. Owing to the long transients that emerge in wide localized devices, the application of this technique to the precise measurement of conductance in systems over the diffusive-localized crossover has not yet been realized.

Our method explicitly includes the leads in the Hilbert space of the simulation and captures the temporal profile of the current as an electric field is applied adiabatically inside the sample. The conductance is extracted from the developed non-equilibrium quasi-steady state. In order to extend this quasi-steady state and allow for a more accurate reading without any additional computational effort, a spatial modulation is applied to the leads' hoppings that closely mimics the semi-infinite lead limit. The expectation value of the current is obtained by resorting to a stochastic evaluation of the trace, replacing a sum over local currents by an average over expectation values of random vectors on cross-sectional currents. The numerical complexity thus becomes linear in the cross-sectional width, rather than quadratic, and the trade-off between the error bar and numerical complexity proves beneficial for wide (width > 1000 unit cells) samples. The time evolution of the system is performed via a Chebyshev expansion of the density matrix and the time evolution operators, leveraging the sparsity of the real-space Hamiltonian while being numerically exact and scaling linearly with the size of the Hilbert space both in time and space complexity.

This new approach is the result of several developments in real-space simulation methods and offers a clear picture of transport in two-dimensional disordered systems. It is easily generalizable to more complex models and geometries. Additionally it can also be used to compute several other quantities, such as local charge and spin density, as time-dependent quantities in the presence of both resonant and non-resonant scatterers. An immediate extension of the present work is the study of the different transport regimes in disordered nanowires, in which a metal-to-insulator phase transition and a stable diffusive metal phase are known to exist [8]. In the diffusive metal phase the conductivity obtained from the conductance's scaling behaviour should exactly match the one obtained using Kubo formalism. Even though we have focused exclusively on the non-equilibrium quasi-steady state, the transient regime is also accessible and should provide valuable information about the systems being studied.

# Acknowledgments

**Funding information** This work was supported by Fundação para a Ciência e a Tecnologia (FCT, Portugal) in the framework of the Strategic Funding UIDB/04650/2020. Further support from Fundação para a Ciência e a Tecnologia (FCT, Portugal) through Projects No. EXPL/FISMAC/0953/2021 (J.M.A.P.), 2022.15885.CPCA, 2023.11029.CPCA and Grant. No. 2023.02155.BD (J.M.A.P) are acknowledged. S.M.J. further acknowledges funding from the Royal Society through a Royal Society University Research Fellowship URF\R\191004 and funding from the EPSRC programme grant EP/W017075/1.

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
