# Peer review of "Unambiguous Simulation of Diffusive Charge Transport in Disordered Nanoribbons"

_SciPost Physics, doi:SciPost Phys. 17, 149 (2024)_

## Round 1 · Referee Report · Anonymous (Referee 1) · 2024-8-16

Strengths

1- A new method to compute quantum transport in 2D materials. This makes a positive contribution to a currently very active field of research into the electronic properties of nanomaterials. 2- The manuscript is well written and the numerical method is well presented.

Weaknesses

1- Perhaps the link with experimental results could be more detailed and explicit in the manuscript.

Report

This manuscript proposes a new numerical method for calculating the conductance in a 2D systems in the presence of an Anderson defects. It is an extension to a ribbon of variable width of the work proposed for a 1D chain. in PRB 101, 104203 (2020) Ref. [68]. That method allows them to simulate the diffusive transport regime expected in 2D systems of size between the mean-free path and localization length.
The work is well presented and appropriate in terms of the description of the numerical method used and the quality of the calculations carried out. It seems to me that these results are new and interesting and deserve to be published. However, I have some comments/questions that the authors could take into account to improve the manuscript.

Requested changes

1- Why this new method? What is really new or advantageous, that cannot be calculated using other methods? In particular those, method based on Kubo-Greenwood approach described in reviews Refs.13 and 25 and in other references cited in the manuscript. It seems to me that other numerical methods can also capture the diffusivity plateau in 2D systems. Moreover, I am surprised by the results shown in figure 9(b) obtained in a Kubo-Greenwood method which is not detailed in this manuscript. The authors conclude that their method does a better job of capturing the effect of geometry, but can they comment a little on why Kubo-Greenwood calculation does not work. 2- The static disorder studied is of the Anderson disorder type. This is entirely appropriate for simulating the effect of a random distribution of non-resonant scatterers. In 2D materials, resonant scatterers such as vacancies or adsorbed atoms are also very important from an applications point of view. Can the method used be applied in this case? 3- The last sentence of section 2.2 is: “From here on forward, the presented timedependent results were computed with f(t) = 1 − e(− τt), where τ is an adiabatic parameter.” Can you explain what this adiabatic parameter is and how it affects the results?

Recommendation

Publish (meets expectations and criteria for this Journal)

  • validity: high
  • significance: good
  • originality: good
  • clarity: high
  • formatting: excellent
  • grammar: -

Author:  Henrique Veiga  on 2024-09-18  [id 4788]

(in reply to Report 1 on 2024-08-16)

We thank the referee for their interest in our work and for recommending publication in the SciPost Physics. Regarding the questions raised in the report, we give below a point-by-point answer including, when applied, an explicit reference to related changes that were made to the new version of the manuscript.

(1) It is true that there are other methods (such as Kubo formulas) that enable to capture the conductivity in the diffusive regime. However, we defend that such methods implicitly assume that the transport is diffusive from the start and, therefore, do not generally allow for the observation of all transport regimes (ballistic, diffusive and localized) within a common framework. This point is precisely the one that we are addressing in this work, i.e., to show that a transmission based method such as ours can observe local and non-local transport regimes in two-dimensional system, matching the remaining methods whenever the scaling of the conductance hints the emergence of a local transport regime.

(2) To change from static disorder to resonant scatterers, one only needs to change the Hamiltonian of the sample. This method is concerned with the density of states of the leads. Therefore, it can be employed in an analogous fashion to study transport in systems with resonant scatterers. From a technical point of view, the real-space approach presented in this study is similar to the one currently available in KITE. As in this project, this newly introduced numerical method is flexible with respect to different implementations of disorder in real space. In the revised version of the manuscript, a sentence in the conclusion was added to indicate the flexibility of the presented method.

(3) Central to this novel approach is the unitary time-evolution of an initially thermalized quantum state. The beginning of this time-evolution is marked by the sudden connection of a uniform electric field through the disordered sample. In doing so, oscillations arise around the quasi-steady state. If the combination of the geometry and disorder strength is such that the system is close to the localized phase, the value of the Landauer current can become comparable to the amplitude of the oscillations. This creates difficulties in properly computing the Landauer current from the time average of the quasi-steady state’s plateau. A solution to this problem is to slowly connect this uniform electric field. In practice, we alter the time-dependence of the biasing potential by introducing the function f(t), whose exponential timescale is the adiabatic parameter. As shown in Eq.22 of the submitted manuscript, this approach is equivalent to performing a moving average of the time-dependent current with a kernel, which is the time derivative of f(t). Throughout the paper the adiabatic parameter, tau, scaled with the total simulation time, T, as tau = T / 18. In the revised version of the manuscript, we have included further motivation for this approach at the end of Section 2.2.

Attached to this reply we include a version of the resubmitted manuscript with the changed text marked in blue font.

Given the referee’s positive feedback and the adjustments made to the text in order to clarify the points raised, we are confident that this version of the manuscript is now ready for publication.

Best Regards,
Henrique Veiga (on behalf of all the authors)

Attachment:

resubmission_changes_Og4Q7kc.pdf

---

## Round 1 · Referee Report · Anonymous (Referee 2) · 2024-8-28

Strengths

Very detailed and relatively easy to follow;

In depth analysis of the onset of the scale invariant diffusive regime;

More in the text.

Weaknesses

The discussion of the role of the ratio xi/L (localization length/sample size) for the diffusion (Fig.8) is not sufficiently clarified, cf. the text of the report;

In my opinion, both panels of Fig.9 cannot be compared directly: the relation between the both abscissas is not clear at all;

More in the text.

Report

Dear Editor,

You have asked my opinion on the sent in manuscript authored by Veiga, João, Pinho, Pires, and Parente Lopes, which concerns some aspects of the physics of diffusive charge transport in disordered nanoribbons. According to the authors themselves, the main achievement of their contribution with the approximate numerical approach to the transport in 2D systems proposed in it, is the possibility to controllable distinguish between three expected transport regimes, namely ballistic, diffusive and localized.

The manuscript at it is represents an extended version of the Master of Science thesis of the first author, which can be found in the library of the University of Porto. For this reason it is written in a very detailed manner and can be recommended for publishing simply because of its pedagogic value: An interested reader will find in it a thorough introduction to this research field and a sufficiently well described application manual for the employed techniques.

The main attention in the original part is devoted to the transport studies of 2D nanoribbons in framework of the Landauer formalism. The model presented in Section 2.1 seems to be sufficiently realistic. In particular it contains the disorder in form of the onsite random potential with a box distribution.

The application of the Landauer formalism to 2D systems is the matter of a discussion lasting already for several decades with personal opinions ranging from total denial of any predictability
to absolute acceptance. Clear is that while the formalism works very well for 1D case, the application in 2D case relies on approximations.

These are introduced and discussed in and around Section 3 and tried on a 1D toy model with
subsequent extension to a quasi-2D systems. While the results for 1D model presented graphically in Fig.5 seem to point to a consistent picture, with the numerical instabilities showing themselves only at very large times, it is still not quite clear how good or bad do these results compare to the known ones. I presume that there should be a vast knowledge accumulated in the field, so a comparison to some sources would be very good.

In Section 4 the emergence of the diffusive regime is discussed. The results are presented in Fig.8, which shows the dimensionless conductance of the quasi-2D disordered ribbon as function of the system size for fixed disorder strength and three different Fermi energies. The onset of the diffusive regime is observed only at half-filling, i.e. for the Fermi energy adjusted to zero and importantly either in thermodynamic limit (system size to infinity), or in the regime with infinite localization length xi, cf. Fig 8c. I would expect a more thorough discussion of this point. For instance, is it possible to model a finite size system with the disorder chosen such that the localization length xi is much larger than the system size? Would one then see the onset of the diffusive regime?

The authors argue that the ultimate test for the correctness of their approximations and approach schemes comes from the comparison with the results obtained from the static Kubo-Greenwood formula in Eq.(49) and shown in Fig.9. Although their talk here about a "direct comparison", I have my difficulties with linking the respective lengths to each other. The Kubo-Greenwood formula is not subtle with respect to the spatial dimension of the system, the trace simply becomes the integral over the Brillouin zone. What is the relation between the respective lengths and how such length appears in the case of the Kubo-Greenwood formula if the evaluation is performed over an infinite sample? At half filling the system lacks any specific intrinsic length.

To conclude: Although I generally tend to take a positive stand towards the publication recommendation of this nice work, I still have some of the reservation. If the authors respond satisfactory to my objections I would be happy to do just so.

Requested changes

In the text

Recommendation

Ask for minor revision

  • validity: high
  • significance: high
  • originality: good
  • clarity: high
  • formatting: good
  • grammar: good

Author:  Henrique Veiga  on 2024-09-18  [id 4787]

(in reply to Report 2 on 2024-08-28)

We thank the referee for their interest in our work and the constructive feedback which helped us improve on the previous version of the manuscript. In following, we reply to all the points raised in the report pointing to the changes done in the manuscript when applied.

(1) The referee is right in pointing out that there is a current discussion regarding the validity of the Landauer formula for two-dimensional samples and leads displaying decoherence mechanisms. In this paper, the diffusive regime is studied under the introduction of disorder. Because disorder is an elastic process, coherence is maintained across the entire sample.

We take the opportunity to also share with the referee that, indeed, a very interesting point that is demonstrated in this work (and already analyzed in the one-dimensional context by some of us [PRB 101, 104203 (2020)]) is that a steady-state regime of transport matching the Landauer current is naturally reached in a system that evolves unitarily. In other words, this stabilization is not conditional on the existence of inelastic relaxation.

(2) As explained in the introduction, this work aims to provide a clear cut observation of a diffusive transport regime across a two-dimensional disordered nanoribbon in the absence of inelastic processes (be they modeled or phenomenologically included). In practice, this means that the authors aimed to observe a sample’s conductance that grows proportionally with the width/length ratio for several orders of magnitude in both system dimensions. For narrow nanoribbons this can be done directly by using RGF methods (e.g. see [New J. Phys. 16, 063065 (2014)]) and this comparison is made in Figure 5 (as the referee points out) but also in Figure 6 (the normalization I_Land is calculated using the RGF method). In the new version of the manuscript this information was explicitly added to the caption.

Note that when the system becomes wider, the RGF methods become computationally impractical and the time-evolution method presented in our manuscript becomes a better alternative. In this regime, there is no comparison that can be done.

(3) As implied by the referee, the definition of localization length in these systems is a subtle matter which may justify a more clear explanation. In this work we are considering disordered nanoribbons for which one can define a finite localization length in the xx direction which depends parametrically on the ribbon’s width. The existence of this scale can be observed by looking at the exponential scaling of the dimensionless conductance as the nanoribbon is made bigger in the xx direction. Moreover, this length is expected to converge to a finite value in the limit of very wide ribbons as shown in the earlier work of Abrahams et al [PRL 42, 673 (1979)].

It is important to remark that, in the absence of inelastic processes, the diffusive regime can only be observed if the localization length exceeds the system size (whilst the elastic mean free path remains much smaller). Otherwise, the transport regime would be localized and the dimensionless conductance would not be size independent. Hence, we were interested precisely in simulating systems that are smaller than their localization length but without dealing with the discreteness of the energy spectrum that arises when treating finite systems. In this work, the decoupling of both effects is achieved by simulating a disordered sample placed between two ideal leads that provide a continuous energy spectrum to the whole system but i) do not generate any further localization of the single electron states (because they are not disordered), and ii) do not contribute to the conductance since there is no electric field applied there. So, answering the referee’s last question: Yes, we are indeed simulating transport through a system that is much smaller than its localization length and this is essential to observe. However, “system size” here refers to the size of the disordered central patch of the mesoscopic device and not the whole device including the leads.

(4) We are sorry that the current text in the manuscript was unclear and implied that we use a comparison between Landauer and Kubo/Greenwood as a way to validate the correctness of our method. Quite oppositely, we show that our method is able to accurately replicate the Landauer current (i.e., coherent quantum transmission) independently of the sample size or the strength of disorder. Then, in theory, the results obtained in this way should correspond to the true quantum dynamics of the many electron system so long as we are not considering the presence of any inelastic decoherence mechanisms or electron-electron interactions.

An alternative to these methods based on quantum transmission, is to directly calculate the conductivity formulated as a Kubo formula. As referred in the introduction, such methods make implicit assumptions on the nature of the transport in the system, namely that it is local and allows for the definition of a bulk conductivity. This is not always true in the quantum world and these methods famously fall short of correctly describing localization even in 1D systems (see Czycholl and Kramer [Z. Physik 39, 193 (1980)]. Furthermore, their practical implementation in numerical calculations involves the use of phenomenological inelastic parameters that may significantly affect the results for dc-transport (see Nikolic [PRB 64, 165303 (2001)]). Notwithstanding, if transport is truly diffusive the Kubo approach is expected to yield the exact same results as the transmission approaches which justifies the comparison made between the two methods at the end of the manuscript. Also, we believe that the two results, while very close, do not exactly match simply because the diffusive regime does not strictly exist in two/dimensions being just a broad crossover regime connecting a ballistic to a localized phase whenever the sample is finite.

Some of us further explored this point in a separate publication [João et al. arXiv:2408.16611]. We showed that the Kubo formula is also able to extract the conductance out of a two-terminal system, obtaining perfect agreement with the Landauer and Kubo approaches in the ballistic regime. In this work, we were unable to probe the diffusive regime due to the poor scaling of the algorithm used, but it corroborated the hypothesis that the mismatch between bulk Kubo and Landauer lies in the underlying assumptions about geometry and locality of transport, rather than the formalism itself.

In order to make this point clearer, we have changed the end of section 4 to highlight the above-mentioned differences between the quantum transmission and the Kubo based approaches to quantum transport calculations. Attached to this reply we include a version of the resubmitted manuscript with the changed text marked in blue font.

By this point, we believe to have addressed all the points raised by the referee and improved the submitted manuscript. We kindly ask that the above replies are taken in due consideration and that the new version of the manuscript is now ready for publication.

Best Regards,
Henrique Veiga (on behalf of all the authors)

Attachment:

resubmission_changes.pdf

---

## Round 2 · Referee Report · Anonymous (Referee 2) · 2024-9-26

Strengths

A nice detailed and accessible presentation of the formalisms and results

Weaknesses

I am still troubled with the message of Fig.9. It is difficult to compare the respective lengths, therefore the similarity of the graphs may well be just occasional.

Report

In this second version the manuscript has improved quite a bit. After having read the Authors responses and the modified manuscript, the key ideas of this nice work became clearer to me.

The manuscript has been thoroughly modified. As I pointed out in my first report, the very detailed presentation of the main formalism and results make it very accessible to an interested reader.

Although I still find some of the passages in the text vague, I think that I can consider the paper as am important mark stone the authors reached and certainly worth of being accepted by SciPost.

It is left to the authors to resolve the nuances in the subsequent work.

Requested changes

Publish as is.

Recommendation

Publish (meets expectations and criteria for this Journal)

---

## Round 2 · Referee Report · Anonymous (Referee 1) · 2024-10-8

Report

The revised version of the manuscript includes some changes that I think improve it. Although I do not entirely agree with the response to my first comment (Kubo-type methods do not always assume a diffusive regime), I find the authors' responses overall convincing. I think the manuscript deserves to be published as it stands.

Recommendation

Publish (easily meets expectations and criteria for this Journal; among top 50%)

---

## Round 2 · List of Changes

(1) The last paragraph of Section 2.2 was updated, so that it includes more information about the adiabatic switching;
(2) A sentence was added to the captions of figures 5 and 6, to clarify that the normalization was computed using a recursive Green’s function algorithm;
(3) Reformulation of the last three paragraphs of Section 4, regarding the comparison with the Kubo-Greenwood formula;
(4) An example of a direct extension of the presented work was added to the last paragraph of the Conclusion;

---

## Editorial Decision

published